# TASEPy: a Python-based package to iteratively solve the inhomogeneous exclusion process

Luca Ciandrini[1,2], Richmond L. Crisostomo[1,3] and Juraj Szavits-Nossan[4*]

**1** Centre de Biologie Structurale (CBS), Univ Montpellier, CNRS, INSERM, Montpellier, France
**2** Institut Universitaire de France (IUF)
**3** Quantitative Biology (qbio) Program, Master Biologie Santé, Faculty of Science, University of Montpellier, France
**4** School of Biological Sciences, University of Edinburgh, Edinburgh EH9 3JH, United Kingdom

* Juraj.Szavits.Nossan@ed.ac.uk

October 31, 2023

## Abstract

The totally asymmetric simple exclusion process (TASEP) is a paradigmatic lattice model for one-dimensional particle transport subject to excluded-volume interactions. Solving the inhomogeneous TASEP in which particles' hopping rate vary across the lattice is a long-standing problem. In recent years, a power series approximation (PSA) has been developed to tackle this problem, however no computer algorithm currently exists that implements this approximation. This paper addresses this issue by providing a Python-based package `TASEPy` that finds the steady state solution of the inhomogeneous TASEP for any set of hopping rates using the PSA truncated at a user-defined order.

## 1   Introduction

The exclusion process is a paradigmatic lattice model for one-dimensional transport subject to excluded-volume interactions. It was introduced as a model for mRNA translation in 1968 [1, 2], and independently in 1970 as a generalization of the lattice random walk to include multiple particles with excluded-volume interactions [3], when it was named the exclusion process. The exclusion process rose to prominence in the 1990s due to its connection to a variety of physical phenomena, including boundary-induced phase transitions [4], vehicular traffic [5], quantum spin chains [6] and surface growth [7]. In nonequilibrium statistical physics, the totally asymmetric simple exclusion process (TASEP) in which particles move unidirectionally and with uniform hopping rates (the homogeneous TASEP) is one of very few models whose nonequilibrium steady-state solution is known exactly [8–10]. The steady-state solution of the homogeneous TASEP via matrix-product *Ansatz* has inspired a large body of research on nonequilibrium steady states [11]. Its large deviation properties [12] have played an important role in the development of the macroscopic fluctuation theory [13] describing coarse-grained fluctuations in driven diffusive systems [14]. Applications of the TASEP in biology, other than mRNA translation [15–18], include DNA transcription [19–23] and intracellular transport by molecular motors [24, 25].

In the context of mRNA translation, the TASEP captures stochastic motion of individual ribosomes on the mRNA, including excluded-volume (steric) interactions between ribosomes that may cause traffic jams. Ribosome speed along the mRNA is non-uniform, which has been linked to variations in the availability of the transfer RNA (tRNA) molecules delivering the correct amino acid to the ribosome [26]. Most amino acids are encoded by two to six (synonymous) codons, whose frequency of usage is non-random [27], which is known as the codon usage bias. In biotechnology, codon (sequence) optimization by replacing rare with frequent codons has become an important tool to increase the production of proteins that are non-native to their host cell, with some studies reporting up to 1000-fold increase in protein levels [28]. These successes are seemingly in contrast to numerous studies demonstrating that translation is rate-limited by initiation [29–32], raising an opportunity for the TASEP to explore theoretically the effect of variable codon speed on the protein production rate.

Variable ribosome speed can be modelled by the TASEP with inhomogeneous hopping rates that are fixed to each lattice site, which we refer to as the inhomogeneous TASEP (other names that circulate in the literature are the TASEP with site-wise disorder and the disordered TASEP). In stark contrast to the homogeneous TASEP for which many exact results are known, solving the inhomogeneous TASEP is a challenging problem, even if all but one lattice sites have the same hopping rate [33–35]. One option is to employ the mean-field approximation, in which correlations between neighbouring particles are ignored. This approximation leads to a set of nonlinear equations for the local particle densities at each lattice position that must be solved

numerically [36,37]. If the hopping rates are slowly varying along the lattice and the number of lattice sites is large, then a hydrodynamic (coarse-grained) limit of the TASEP is justified, yielding simple analytical results for the local density [38]. Finally, if the TASEP has only few sites with slow hopping rates, then a combination of the mean-field approximation and the exact solution of the homogeneous TASEP can be used [34,39,40].

The other option, which accounts for correlations between particles, is the power series approximation (PSA) [41–43]. This approximation is based on the formal series expansion of the steady-state solution, with the initiation rate (at which new particles are added to the lattice) being the expansion variable. The coefficients in this expansion can be computed exactly, whereas the approximation comes from the truncation of the series at some given order. Since initiation is typically the rate-limiting step in translation, only the first few terms are needed to obtain accurate results [42]. Consequently, the power series approximation is expected to be considerably faster than the stochastic simulation algorithm (the Gillespie algorithm) [44]. This advantage has proved instrumental for solving the inverse problem of inferring variable ribosome speed from experimental data obtained by ribosome profiling [45]. However, computer implementation of the power series approximation has so far been limited to the first few orders, preventing its wider use.

In this paper, we close this gap by providing a computer algorithm that finds the steady state of the inhomogeneous TASEP for arbitrary hopping rates using the power series truncated at an arbitrary, user-defined order. After introducing the model, we summarize the power series approximation [41–43] and present an iterative method to solve it for any order. Finally, we implement this method in a Python code, which we package as the TASEPy module. We provide detailed instructions how to use it, and test its correctness using exact results for small systems and stochastic simulations for large systems. The code is available in a GitHub repository [46] under the MIT licence.

## 2  The inhomogeneous TASEP

### 2.1  Definition of the model

We model a driven gas of particles advancing on a unidimensional discrete lattice consisting of $L$ sites, labelled from 1 to $L$. As we are motivated by the mRNA translation process [1], we assume that each particle occupies $\ell$ sites. In order to locate each particle on the lattice, we arbitrarily chose a site of the particle that we will call the "tracking site" (identical for all particles); we can then identify the position of a particle on the lattice by the position of its 'tracking' site. For instance, in Fig. 1 each particle has size $\ell = 3$ and the tracking site is the middle one. For ribosomes, $\ell \approx 10$ and the tracking site is approximately 5 lattice sites from the ribosome's trailing end. However, the procedure that we describe is general, meaning that it can be applied to any value of $\ell$, and it does not depend on the position of the particle's tracking site [2,15].

The model is illustrated in Fig. 1. Particles enter the lattice with their tracking site (highlighted with a black dot) on site 1 with a probability per unit time $\alpha$, provided that no other particles interfere with the binding of that particle. In other words, the tracking site of the following particle should be at least $\ell + 1$ sites downstream. In the following, we will often identify the position of a particle with the position of its tracking site. In the bulk, a particle moves from site $i$ to site $i + 1$ with a rate $\omega_i$, provided that there is no particle on site $i + \ell$. On the last site $L$, particles exit the lattice with a rate $\beta$.

For each site $i = 1, \dots, L$ we define the corresponding particle occupancy number $\tau_i \in$

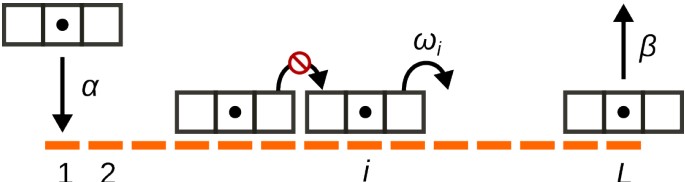

Figure 1: Sketch of the TASEP for $\ell = 3$. A black dot denotes the tracking site. A particle is injected with its tracking site on the first lattice site with rate $\alpha$. The position of the tracking site then moves from site $i$ to $i+1$ with rate $\omega_i$, and when it reaches the last site, the particle is ejected from the lattice with rate $\beta$. At each step, the dynamics has to respect the exclusion rules explained in the text.

$\{0, 1\}$,

$$
\tau_i = \begin{cases} 1 & \text{if site } i \text{ is occupied by a particle's tracking site,} \\ 0 & \text{otherwise.} \end{cases} \tag{1}
$$

These numbers determine the configuration of the system denoted by $C = \{\tau_1, \ldots, \tau_L\}$. Using this notation, kinetic steps of the driven lattice gas can be summarized as:

$$
\text{(initiation): } \tau_1 = 0 \xrightarrow{\alpha} 1 \text{ if } \tau_1 = \cdots = \tau_\ell = 0 \tag{2a}
$$

$$
\text{(elongation): } [\tau_i, \tau_{i+1}] = [1, 0] \xrightarrow{\omega_i} [0, 1] \text{ if } \tau_{i+\ell} = 0, \quad i = 1, \ldots, L - \ell \tag{2b}
$$

$$
[\tau_i, \tau_{i+1}] = [1, 0] \xrightarrow{\omega_i} [0, 1], \quad i = L - \ell + 1, \ldots, L \tag{2c}
$$

$$
\text{(termination): } \tau_L = 1 \xrightarrow{\beta} 0. \tag{2d}
$$

Eqs. (2a)-(2d) define the totally asymmetric simple exclusion process first proposed by Mac-Donald, Gibbs and Pipkin as a model of mRNA translation [1].

## 2.2 Particle current and particle density

Our main goal is to compute the particle current $J$ and particle densities $\rho_i$ on each site $i$, in the steady state. As we do not consider particle binding and unbinding inside the lattice, the current $J$ in the steady state is conserved across the lattice, and we can write:

$$
J = \alpha \left\langle \prod_{i=1}^{\ell} (1 - \tau_i) \right\rangle = \beta \langle \tau_L \rangle \tag{3}
$$

where the brackets $\langle \ldots \rangle$ denote an ensemble average with respect to the steady-state probability $P(C)$. As the steady-state current is conserved along the lattice, the equation above can be understood as equating the entry current (injection rate $\alpha$ multiplied by the probability that the first $\ell$ sites are empty), and the exit current on the last site (termination rate $\beta$ multiplied by the probability of occupancy of the last site). The local particle densities $\rho_i$ (the probability of finding a particle tracking site on site $i$), and the average lattice density $\rho$ are defined as

$$
\rho_i = \langle \tau_i \rangle, \tag{4a}
$$

$$
\rho = \frac{1}{L} \sum_{i=1}^{L} \rho_i. \tag{4b}
$$

## 2.3  Steady-state master equation

The steady-state probability $P(C)$ satisfies the master equation,

$$0 = \sum_{C'} W(C' \to C)P(C') - \sum_{C'} W(C \to C')P(C), \tag{5}$$

where $W(C \to C')$ denotes the rate of transition from configuration $C$ to $C'$. To be specific, we adopt an alternative notation for $C$ that specifies the positions $x_m$ of each particle $m$ on the lattice (i.e. the position of its tracking site),

$$C = \{\tau_1, \ldots, \tau_i, \ldots, \tau_L\} = \{x_1, \ldots, x_m, \ldots, x_N\}, \quad N = \sum_{i=1}^{L} \tau_i, \quad \tau_{x_m} = 1, \, m = 1, \ldots, N \tag{6}$$

In other words, $N$ is the number of particles in configuration $C$, and $x_1, \ldots, x_N$ are the positions of particles in $C$, where $x_1$ is the position of the leftmost and $x_N$ the position of the rightmost particle. We use index $i$ to indicate the lattice site, index $m$ to indicate the $m$th particle, while $n$ is reserved to the order of the power series expansion. The particle positions $x_1, \ldots, x_N$ must respect the excluded-volume interactions,

$$x_m + \ell \leq x_{m+1}, \quad m = 1, \ldots, L - 1. \tag{7}$$

To simplify the notation, we define $\omega_0 = \alpha$ and $\omega_L = \beta$. The steady-state master equation, Eq. (5), can be written as

$$\underbrace{e(C)P(\{x_1, \ldots, x_N\})}_{\text{going out of } C} = \underbrace{\sum_{m=1}^{N} \omega_{x_m-1} \mathbb{E}^m P(\{x_1, \ldots, x_N\}) + \omega_L P(\{x_1, \ldots, x_N, L\}) \mathbb{1}_{x_N \leq L-\ell}}_{\text{going into } C}, \tag{8}$$

where $e(C)$ is the exit rate from configuration $C$,

$$e(C) = \sum_{C'} W(C \to C') = \omega_0 \mathbb{1}_{x_1 > \ell} + \sum_{m=1}^{N-1} \omega_{x_m} \mathbb{1}_{x_{m+1}-x_m > \ell} + \omega_{x_N}, \tag{9}$$

$\mathbb{1}_A$ is the indicator function that is equal to 1 if the condition $A$ is true and is 0 otherwise, and $\mathbb{E}^m$ is the ladder operator that moves the $m$th particle one lattice point to the left, provided the move is allowed by the excluded-volume interactions. For $x_m > 1$, the ladder operator $\mathbb{E}^m$ is defined as

$$\mathbb{E}^m P(\{x_1, \ldots, x_N\}) = \begin{cases} P(\{x_1, \ldots, x_m - 1, \ldots, x_N\}) & x_m - x_{m-1} > \ell, \\ 0 & x_m - x_{m-1} = \ell. \end{cases} \tag{10}$$

For $x_1 = 1$, the ladder operator $\mathbb{E}^1$ is defined as

$$\mathbb{E}^1 P(\{1, x_2, \ldots, x_N\}) = \begin{cases} P(\{x_2, \ldots, x_N\}) & N > 1, \\ P(\emptyset) & N = 1, \end{cases} \tag{11}$$

where $\emptyset$ denotes the empty lattice.

In summary, to find the master equation for a given configuration $C$ of particles, we compute the exit rate $e(C)$ from that configuration [the left-hand side of Eq. (8)], and then check all particles that can be moved one step to the left—that determines the configurations $C'$ that $C$ can be entered from [the right-hand side of Eq. (8)]. For example, consider a system with

$\ell = 10$ and $L = 100$, and a configuration with three particles $C = \{x_1, x_2, x_3\} = \{1, 11, 30\}$. The master equation for $P(1, 11, 30)$ reads,

$$(\omega_{11} + \omega_{30})P(1, 11, 30) = \omega_0 P(11, 30) + \omega_{29}P(1, 11, 29) + \omega_{100}P(1, 11, 30, 100). \quad (12)$$

The first particle at $x_1 = 1$ cannot move because the second particle is at $x_2 = x_1 + \ell = 11$. This simple way of generating the master equation for each configuration is useful for the power series approximation discussed in the next section.

## 3 Power series approximation (PSA)

### 3.1 General results

The power series approximation (PSA), as previously formulated in the Refs. [41–43, 47, 48], represents $P(C)$ as a power series expansion in the initiation rate $\alpha$,

$$P(C) = \sum_{n=0}^{\infty} c_n(C)\alpha^n. \quad (13)$$

The unknown coefficients $c_n(C)$ depend on both the particle configuration $C$ and the rates $\omega_1, \ldots, \omega_L$. Considering that the sum of all $P(C)$ must equal to 1, it directly follows that

$$\sum_C c_n(C) = \begin{cases} 1, & n = 0 \\ 0 & n \geq 1. \end{cases} \quad (14)$$

While it is possible to expand $P(C)$ in other rates, we choose to expand in the initiation rate $\alpha$ as we are mainly motivated by mRNA translation for which we expect the translation initiation rate to be much smaller than any other rate. To illustrate, the median value of $\alpha$ estimated for the *S. cerevisiae* genome is an order of magnitude smaller than any of the elongation rates [31]. This let us approximate the series expansion $P(C)$ in Eq. (13) using the first $K$ terms

$$P(C) \approx c_0(C) + c_1(C)\alpha + \cdots + c_K(C)\alpha^K. \quad (15)$$

We emphasize that, although Eq. (13) is exact, we may introduce notable errors by truncating the series expansion to a limited number of terms as done in Eq. (15). Considering an initiation rate comparable to other rates may thus result in non-physical values of $P(C) < 0$ or $P(C) > 1$. In particular, if for a specific choice of $\alpha$ the approximation fails, it becomes necessary to include higher-order terms in Eq. (15). Later in this work, we will provide criteria to establish the reliability of the PSA's results.

To compute the steady-state probabilities $P(C)$ and consequently calculate particle currents and densities, we first need to determine the coefficients $c_n(C)$. We insert Eq. (13) into Eq. (5) and gather terms involving $\alpha^n$. The sum of these terms equals to zero, since the left-hand side of Eq. (5) equals zero. Next, we differentiate between cases where $W(C \rightarrow C') = \alpha$ (resulting in terms of the order $\alpha^{n+1}$ when multiplied by $P(C)$) and cases where $W(C \rightarrow C') \neq \alpha$ (yielding terms of the order $\alpha^n$). Using an indicator function $I_{C,C'}$,

$$I_{C,C'} = \begin{cases} 1 & C \rightarrow C' \text{ corresponds to initiation} \\ 0 & \text{otherwise}, \end{cases} \quad (16)$$

we write $W(C \rightarrow C')$ as

$$W(C \rightarrow C') = \alpha I_{C,C'} + W(C \rightarrow C')(1 - I_{C,C'}) = \alpha I_{C,C'} + W_0(C \rightarrow C') \quad (17)$$

where $W_0(C \to C') = (1 - I_{C,C'})W(C \to C')$. Inserting $P(C)$ from (13) into (5) we obtain

$$\sum_{C'} \left[ I_{C',C} \sum_{n=1}^{\infty} c_{n-1}(C')\alpha^n + W_0(C' \to C) \sum_{n=0}^{\infty} c_n(C')\alpha^n \right] =$$
$$\sum_{C'} \left[ I_{C,C'} \sum_{n=1}^{\infty} c_{n-1}(C)\alpha^n + W_0(C \to C') \sum_{n=0}^{\infty} c_n(C)\alpha^n \right]. \quad (18)$$

After gathering all terms containing $\alpha^n$ and equating their sum to 0, we obtain the following equation for $c_n(C)$ for $C \neq \emptyset$

$$c_n(C) = \frac{1}{e_0(C)} \left( \sum_{C'} W_0(C' \to C)c_n(C') + \sum_{C'} c_{n-1}(C')I_{C',C} - c_{n-1}(C) \sum_{C'} I_{C,C'} \right), \quad (19)$$

where $e_0(C)$ is the total exit rate from $C$ excluding initiation

$$e_0(C) = \sum_{C'} W_0(C \to C'). \quad (20)$$

For $C = \emptyset$, we can use Eq. (14) instead, which gives

$$c_n(\emptyset) = \delta_{n,0} - \sum_{C' \neq \emptyset} c_n(C'). \quad (21)$$

The equation (19) applies to $n \geq 1$. For $n = 0$, the equation is simpler and reads,

$$e_0(C)c_0(C) = \sum_{C'} W_0(C' \to C)c_0(C'). \quad (22)$$

We remark that Eq. (22) coincides with the initial master equation when $\alpha$ is set to 0. Since in that case there is no initiation,

$$c_0(C) = \begin{cases} 1, & C = \emptyset \\ 0, & \text{otherwise.} \end{cases} \quad (23)$$

From (23) it follows that all coefficients $c_n(C)$ of order $n$ smaller than the total number of particles $N(C)$ in $C$ will be equal to zero. This is summarized in the condition

$$c_n(C) = 0 \qquad \text{if } n < N = \sum_{i=1}^{L} \tau_i. \quad (24)$$

Mathematically, this result can be derived using the Markov chain tree theorem [49], known in physics as Schnakenberg network theory [50]. Here we omit the details of the derivation, which can be found in Ref. [43] where we proved (24) for the TASEP with particles of size $\ell = 1$. The same arguments hold to the general case studied in this paper. Relations (24) tell us that for each order $n$ we only have to consider lattice configurations with at most $N = n$ particles, and that considerably simplifies the calculation of the coefficients $c_n(C)$. This simplification plays a crucial role in making the power series approximation practical and applicable. In that sense, the power series expansion can be seen as a form of perturbation theory, where each initiation event corresponds to one order of the perturbation theory, starting from the empty lattice.

Applying the power series to Eqs. (3)-(4b), we get the following expressions for the steady-state particle current $J$, local density $\rho_i$ and the average density $\rho$,

$$J = \sum_{n=0}^{\infty} J_n \alpha^{n+1}, \quad J_0 = 1, \quad J_n = c_{n-1}(\emptyset) + \sum_{\substack{C \\ x_1 \geq \ell+1}} c_{n-1}(C), \quad n \geq 1, \tag{25a}$$

$$\rho_i = \sum_{n=0}^{\infty} \rho_{i,n} \alpha^n, \quad \rho_{i,0} = 0, \quad \rho_{i,n} = \sum_{\substack{C \\ \tau_i = 1}} c_n(C), \quad i = 1, \dots, L, \quad n \geq 1, \tag{25b}$$

$$\rho = \frac{1}{L} \sum_{i=1}^{L} \rho_i = \sum_{n=0}^{\infty} \rho_n \alpha^n, \quad \rho_0 = 0, \quad \rho_n = \frac{1}{L} \sum_{i=1}^{L} \sum_{\substack{C \\ \tau_i = 1}} c_n(C), \quad n \geq 1. \tag{25c}$$

Here, index $n$ and the coefficients $J_n$, $\rho_{i,n}$ and $\rho_n$ correspond to the order of the series expansion of $P(C)$ in Eq. (13). Since the current is by definition multiplied by $\alpha$, the 0th order in the series expansion of $P(C)$ contributes to the 1st order in the series expansion of $J$.

## 3.2 Analytical solution for the first-order coefficients

The first order in the power series expansion can be solved analytically. According to Eq. (24), the coefficients $c_1(C)$ are non-zero for configurations having a number of particles smaller than 1, i.e. $C = \{x_1\}$ and $C = \emptyset$. The equation for $c_1(\{x_1\})$ reads,

$$c_1(\{1\}) = \frac{1}{\omega_1}, \quad c_1(\{x_1\}) = \frac{\omega_{x_1-1}}{\omega_{x_1}} c_1(\{x_1 - 1\}), \quad x_1 = 2, \dots, L. \tag{26}$$

This recurrence relation can be easily solved, from which we get that

$$c_1(\{x_1\}) = \frac{1}{\omega_{x_1}}, \quad x_1 = 1, \dots, L. \tag{27}$$

The expression for $c_1(\emptyset)$ follows from Eq. (21),

$$c_1(\emptyset) = -\sum_{x_1=1}^{L} \frac{1}{\omega_{x_1}}. \tag{28}$$

## 3.3 Iterative solution for higher-order coefficients

We can compute all $c_n(C)$ of a given order $n$ recursively by following a natural order, which forms the basis of the code we developed. To see that, we rewrite Eq. (19) for $c_n(C)$ using the notation $C = \{x_1, \dots, x_N\}$, $1 \leq N \leq n$, as previously done for the master equation (8). The pedagogical case $\ell = 1$ is presented in Appendix A, while below we directly derive the equations for the general case $\ell \geq 1$. We obtain

$$c_n(\{x_1, \dots, x_N\}) = \frac{1}{e_0(\{x_1, \dots, x_N\})} \Big( \underbrace{c_{n-1}(\{x_2, \dots, x_N\}) \mathbb{1}_{x_1=1}}_{(a)}$$

$$+ \underbrace{\omega_{x_1-1} \mathbb{E}^1 c_n(\{x_1, \dots, x_N\}) \mathbb{1}_{x_1>1}}_{(b1)} + \underbrace{\sum_{m=2}^{N} \omega_{x_m-1} \mathbb{E}^m c_n(\{x_1, \dots, x_N\})}_{(b2)}$$

$$+ \underbrace{\omega_L c_n(\{x_1, \dots, x_N, L\}) \mathbb{1}_{x_N \leq L-\ell} \mathbb{1}_{n \geq N+1}}_{(c)} - \underbrace{c_{n-1}(\{x_1, \dots, x_N\}) \mathbb{1}_{x_1>\ell} \mathbb{1}_{n-1 \geq N}}_{(d)} \Big), \tag{29}$$

where $e_0(\{x_1, \ldots, x_N\})$ is given by

$$e_0(\{x_1, \ldots, x_N\}) = \sum_{m=1}^{N-1} \omega_{x_m} \mathbb{1}_{x_{m+1} - x_m > \ell} + \omega_{x_N}. \tag{30}$$

Each term on the right-side of Eq. (29) has a simple interpretation, which we explain below:

(a) The first term on the right-hand side of Eq. (29) is a contribution to $c_n(C)$ from the previous order $n-1$, provided the first particle is at site $x_1 = 1$.

(b) The second and third terms account for all configurations that lead to $C$ by moving one particle to the right, provided the move is allowed by excluded-volume interactions. The second term is for the leftmost particle (b1), and the third term is for all other particles (b2).

(c) The fourth term computes the contribution to $c_n(C)$ coming from a configuration that has an extra particle at the last site $L$, provided that $x_N \leq L - \ell$ and $N + 1 \leq n$ because of Eq. (24).

(d) The last term removes the contribution coming from the same configuration but from the previous order $n-1$, provided $x_1 > \ell$. This term is not zero, provided $n - 1 \geq N$.

There is a natural hierarchy of configurations for solving Eq. (29) recursively, such that all coefficients on the right-hand side are known before computing $c_n(C)$ on the left-hand side. Let us assume that we have calculated the values of all non-zero coefficients $c_{n-1}$. We start with a configuration in which $n$ particles are stacked together at positions $x_i = 1 + (i-1)\ell$ for $i = 1, \ldots, n$. We refer to this configuration as a 'stacked' configuration. For now, we assume that $n$ is smaller or equal to the maximum number of particles that can fit onto the lattice, $N_{\max} = \lfloor L/\ell \rfloor + 1$. For the stacked configuration in that case,

$$c_n(\{1, 1+\ell, \ldots, 1+(n-1)\ell\}) = \frac{1}{\omega_{x_n}} c_{n-1}(\{1+\ell, \ldots, 1+(n-1)\ell\}). \tag{31}$$

where $x_n = 1 + (n-1)\ell$ is the position of the rightmost particle. Importantly, the right-hand side depends only on a coefficient of the order of $n-1$, which is known. The next coefficient that can be computed is for a configuration in which the $n$th particle is moved one lattice site to the right ($x_n = 2 + (n-1)\ell$). The coefficient for this configuration depends on the previously computed coefficient ($c_n(\{1, \ldots, 1+(n-1)\ell\})$) and a coefficient that is of the order of $n-1$. This procedure continues for all allowed positions of the $n$th particle, the last being $x_N = L$. The next configuration after that one is obtained by removing the $n$th particle from the lattice.

After we have exhausted all positions of the $n$th particle, we move the $(n-1)$th particle one lattice site to the right, and set the $n$th particle next to it, stacked together. We then leave the $(n-1)$th particle intact while we cycle through all allowed positions of the $n$th particle, including the configuration in which the $n$th particle is removed from the lattice. This procedure is repeated until $(n-1)$th and $n$th particle are both removed from the lattice. We then move $(n-2)$th particle one lattice site to the right and stack the two removed particles next to it, if possible. This procedure is repeated for all particles on the lattice, until we finally reach a configuration in which the first particle is at the last site. The next configuration from there would be an empty lattice, but we can get the coefficient $c_n(\emptyset)$ also by summing all other coefficients of the same order, according to Eq. (14). For $n \geq N_{\max}$, we start from a stacked configuration with $N_{\max}$ particles and cycle through all configurations as before until we reach an empty lattice. In other words, for each $n \geq N_{\max}$ we cycle through all lattice configurations, whereas for $n < N_{\max}$ we cycle through only a subset of configurations because of Eq. (24).

For each order of the PSA $n$, we denote by $S_n$ the set of all configurations which are visited by the iterative procedure explained above. From Eq. (24), it follows that $S_n = \{C \mid N(C) \leq n\}$, where $N(C)$ is the number of particles in $C$. A graphical representation of the iterative procedure for $n = 0, 1$ and $2$ for the TASEP with $L = 4$ and $\ell = 2$ is presented in Fig. 2. In this example, $S_0 = \{\{\}\}$, $S_1 = \{\{1\},\{2\},\{3\},\{4\},\{\}\}$ and $S_2 = \{\{1,3\},\{1,4\},\{1\},\{2,4\},\{2\},\{3\},\{4\},\{\}\}$. For $n \geq N_{\max}$, $S_n$ contains all lattice configurations.

## 4  `TASEPy` **usage instructions**

In this section, we provide a practical implementation of the power series approximation coded into a Python-based package named `TASEPy`. To use this package, copy the `TASEPy.py` file into the working directory and import the following functions:

```
from TASEPy import psa_compute
from TASEPy import total_coeffs
from TASEPy import local_density
from TASEPy import mean_density
from TASEPy import current
```

In the upcoming sections, we will explore the functions used to evaluate densities and currents obtained by the PSA. For a comprehensive step-by-step guide and additional details, please refer to the tutorial *tutorial_TASEPy.ipynb*. Finally, in the last section, we will present benchmarks involving exact results and stochastic simulations.

### 4.1  **Function** `psa_compute`

At the core of TASEPy is the `psa_compute` function. It takes a list containing hopping rates $\omega_1, \ldots, \omega_L$, the maximum order of the series expansion $K$ and the particle size $\ell$ ($\ell = 1$ by default) as input, and returns a two-dimensional list containing coefficients $\rho_{i,n}$ and a one-dimensional list containing coefficients $J_n$, for all $n = 0, \ldots, K$. These coefficients are needed to compute density profiles and particle current in the power-series approximation for any value of $\alpha$, following Eqs. (25a)-(25c). In the following code,

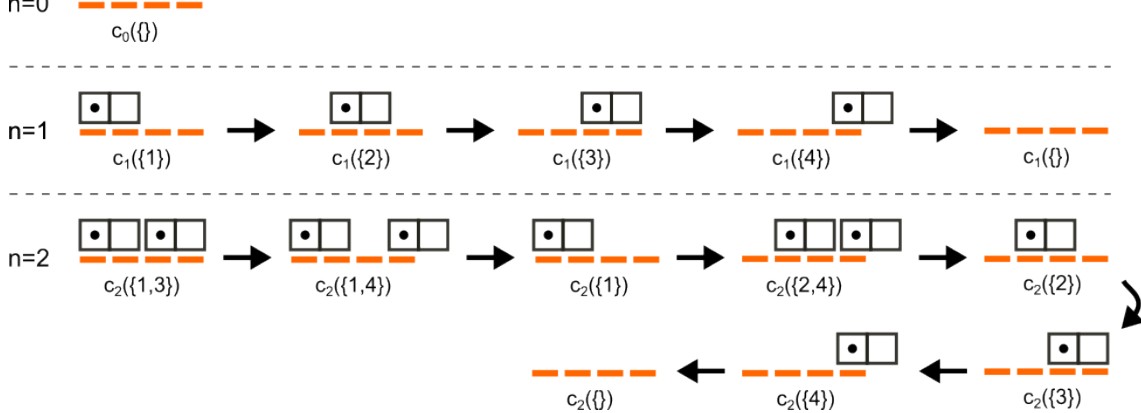

Figure 2: Graphical representation of the iteration procedure for $L = 4$ and $\ell = 2$. For $n > N_{\max} = 2$, the iteration procedure is the same as for $n = 2$. Orders are separated by dashed horizontal lines. Following this procedure, the coefficient $c_n(C)$ of a given configuration $C$ depends only on previously computed coefficients.

```
    rhocoeff, Jcoeff = psa_compute(wlist, K, ll)
```

`wlist` $= [\omega_1, \ldots, \omega_L]$ is the list of hopping rates and `ll` is the particle size $\ell$. The coefficients $\rho_{i,n}$ and $J_n$ are stored in `rhocoeff` and `Jcoeff`, respectively, such that `rhocoeff[i]` is the list $[\rho_{i+1,0}, \ldots, \rho_{i+1,K}]$ and `Jcoeff[i]` $= J_i$.

By default, the function `psa_compute` does not save the coefficients $c_n(X)$. If one is interested in physical quantities other than the local density and current, then it is paramount that the coefficients are saved. This option is activated by setting `save_coeff=True` in the function `psa_compute` (the default value is False if omitted), which saves the coefficients in a file. The name of this file is specified by an optional argument `coeff_file=`'⟨*filename*⟩', e.g. 'test.csv' in the example below:

```
    rhocoeff, Jcoeff = psa_compute(wlist, K, ll, True, 'test.csv')
```

We note that the resulting file may be very large, since the number of coefficients grows exponentially with the system size. We have therefore implemented a function 'total_coeffs' that counts the total number of coefficients that will be stored in the file. It is advised that this function is called before 'psa_compute' when the option 'save_coeffs' is set to True. For example, for $L = 100$, $\ell = 1$ and $K = 4$, the total number of coefficients is close to 4,300,000, which takes about 173 MB, or about 43 bytes per coefficient.

The inner workings of `psa_compute` are the following. For a given order $n$, we generate the initial stacked configuration $X$ as a list of particle positions and compute the coefficient $c_n(X)$ using Eq. (31). This coefficient is then stored in a dictionary `c_n` with key $X$. Using function `next_config()`, we go through all other configurations $X \in S_n$ and use Eq.(29) to compute the corresponding coefficient $c_n(X)$, which is stored in the dictionary `c_n` using key $X$. Due to the hierarchy of configurations explained in Section 3, computing $c_n$ requires only coefficients that have already been stored in the dictionaries `c_n-1` and `c_n`. As we go through all configurations, we compute the density coefficients $\rho_{i,n}$ for each lattice site $i$ using Eq. (25b), and the particle current coefficients $J_n$ using (25a). This process is repeated for all orders $n = 0, \ldots, K$. We remark that, while computing the coefficients of order $n$, `psa_compute` only keeps in memory the dictionaries of order $n-1$ and $n$.

## 4.2  Function `local_density`

Following the execution of `psa_compute`, we can compute the density profile for each order $n$ and for any fixed value of $\alpha$. This is implemented in the function `local_density`,

```
    rho = local_density(rhocoeff, alpha)
```

which takes as input the two-dimensional list `rhocoeff` and the value of the initiation rate $\alpha$, and returns a two-dimensional list containing the density profiles for all orders $0, \ldots, K$. Let us denote by $\rho_i^{(n)}$ the local density truncated at the order of $n$,

$$\rho_i^{(n)} = \sum_{k=0}^{n} \rho_{i,k} \alpha^k. \tag{32}$$

The density profile $[\rho_1^{(n)}, \ldots, \rho_L^{(n)}]$ stored as `rho[n]`. For instance, `rho[2]` is a list containing local particle densities for each lattice site, truncated at the second order.

### 4.3 Function `mean_density`

Once the local density is computed, the mean particle density $\rho = \sum_{i=1}^{L} \rho_i / L$ can be calculated using the function `mean_density`,

```
mean_rho = mean_density(rho)
```

which takes `rho` as input and returns a one-dimensional list $[\rho^{(0)}, \ldots, \rho^{(K)}]$, where $\rho^{(n)} = \sum_{i=1}^{L} \rho_i^{(n)}/L$ for $n = 0, \ldots, K$.

### 4.4 Function `current`

Finally, the particle current $J$ for a given value of $\alpha$ can be computed from Eq.(25a) knowing the current coefficients $J_0, \ldots, J_K$. This is done by the function `current`,

```
J = current(Jcoeff, alpha)
```

which takes the list `Jcoeff` and the value of $\alpha$ as input, and returns the list $[J^{(0)}, \ldots, J^{(K)}]$, where $J^{(n)} = \sum_{k=0}^{n} J_k \alpha^{k+1}$.

This whole procedure is demonstrated in the tutorial file *tutorial_TASEPy.ipynb* for lattice size $L = 100$, particle size $\ell = 1$ and the PSA order $K = 4$, using randomly generated hopping rates in the range between 1 and 10. The resulting density profiles $\rho_i^{(n)}$ for $n = 1, \ldots, 4$ are presented in Fig. 3. In Fig. 4(a) and Fig. 4(b) we show the mean density $\rho^{(n)}$ and particle current $J^{(n)}$ versus $\alpha$, respectively.

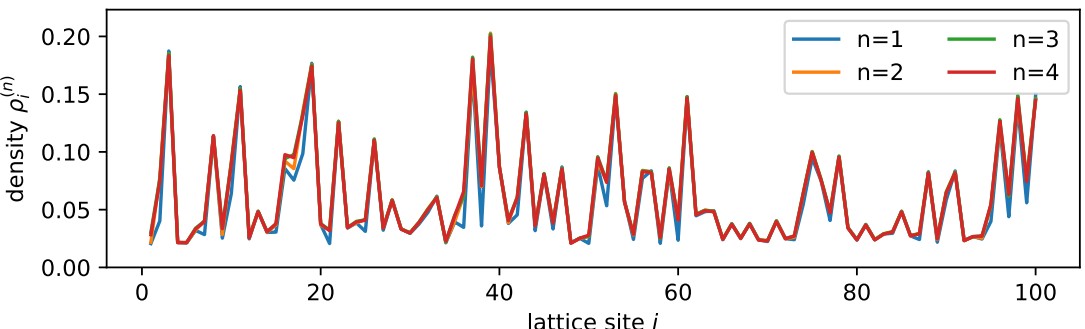

Figure 3: Density profile $\rho_i^{(n)}$ computed using TASEPy at $\alpha = 0.2$ for $L = 100$, $\ell = 1$, $K = 4$ and $n = 1, \ldots, K$. The values of $\omega_1, \ldots, \omega_L$ were selected randomly between 1 and 10, as explained in the tutorial file *tutorial_TASEPy.ipynb*.

Plotting mean density and current versus $\alpha$ is useful for estimating values of $\alpha$ for which the PSA of given order is no longer a good approximation. As $\alpha$ is increased, the highest-order terms in the PSA begin to dominate, leading to values diverging from the exact ones. We know that $0 \leq \rho_i \leq 1$ and $0 \leq J \leq \alpha$, so any value outside these bounds indicates that $\alpha$ is too big. We also expect the local density and current to be non-decreasing in $\alpha$. i.e. that $d\rho_i/d\alpha \geq 0$ and $dJ/d\alpha \geq 0$. Using Eq. (3), we also get that $dJ/d\alpha \leq 1$. These conditions can be easily checked using coefficients stored in `rhocoeff` and `Jcoeff`, as explained in the tutorial file *tutorial_TASEPy.ipynb* where we find the smallest value of $\alpha$ for which any of the conditions above fails. In practice, however, it is best to consider values of $\alpha$ that are much smaller than this value.

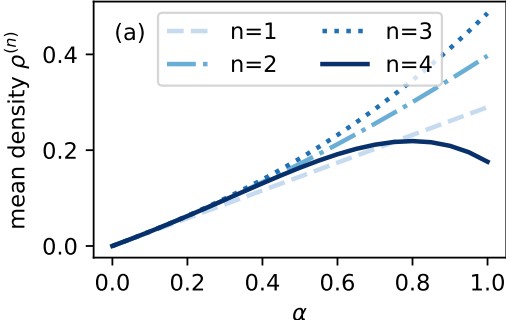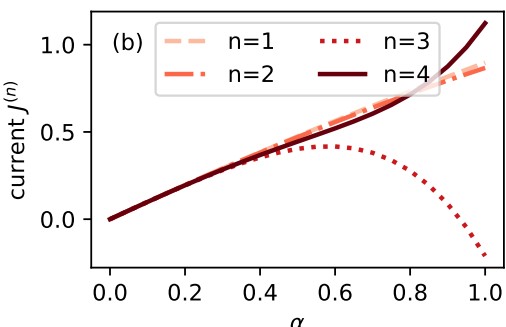

Figure 4: Mean density $\rho^{(n)}$ and current $J^{(n)}$ computed using TASEPy at different values of $\alpha$ for $L = 100$, $\ell = 1$, $K = 4$ and $n = 1, \ldots, K$. The values of $\omega_1, \ldots, \omega_L$ were selected randomly between 1 and 10, as explained in the tutorial file *tutorial_TASEPy.ipynb*.

## 4.5 Benchmarks

The file `benchmarks_TASEPy.ipynb` is a Jupyter notebook comparing the results obtained with TASEPy to symbolic exact calculation for small systems and to stochastic simulations.

Exact results were obtained by solving the stationary master equation $MP = 0$ in Eq. (5) for small lattices, where $M$ is the stochastic transition matrix, and $P$ is the probability vector. This system of equations was solved exactly using Mathematica® to obtain $P$ as a function of the initiation rate $\alpha$, which was kept as a variable. The probability vector was then expanded in $\alpha$ up to the order of $K$, from which the coefficients $\rho_{i,n}$ and $J_n$ were computed for $n = 0, \ldots, K$. The Mathematica® code is available in the repository.

The results were obtained for $L = 4$ and various particle sizes ($\ell = 1, 2$ and $3$) using an arbitrary set of hopping rates $[1.88, 1.52, 1.09, 1.38]$, and were expanded up to the order of $K = 5$. The exact coefficients $\rho_{i,n}$ match those obtained by TASEPy, as we demonstrate below for $\ell = 2$:

```
Exact results (local density):
  site    order 0    order 1    order 2    order 3    order 4    order 5
 ------  ---------  ---------  ---------  ---------  ---------  ---------
      1          0   0.531915   0.149892   0.846462   -1.08538   -4.66958
      2          0   0.657895   0.564896   0.134411   -3.24941    1.19327
      3          0   0.917431   0.176094  -0.902102  -0.936768   0.818141
      4          0   0.724638  -0.385446  -0.108617  -0.613378   0.786505

TASEPy results (local density):
  site    order 0    order 1    order 2    order 3    order 4    order 5
 ------  ---------  ---------  ---------  ---------  ---------  ---------
      1          0   0.531915   0.149892   0.846462   -1.08538   -4.66958
      2          0   0.657895   0.564896   0.134411   -3.24941    1.19327
      3          0   0.917431   0.176094  -0.902102  -0.936768   0.818141
      4          0   0.724638  -0.385446  -0.108617  -0.613378   0.786505
```

Similarly, the coefficients $J_n$ calculated with TASEPy also match the exact ones:

```
Exact results (current):
  order 0    order 1    order 2    order 3    order 4    order 5
 ---------  ---------  ---------  ---------  ---------  ---------
         1   -1.18981  -0.112236    3.57259   -7.65441    7.05124
```

```
TASEPy results (current):
  order 0    order 1    order 2    order 3    order 4    order 5
---------  ---------  ---------  ---------  ---------  ---------
        1   -1.18981  -0.112236    3.57259   -7.65441    7.05124
```

Test results for other particle sizes $\ell = 1$ and 3 are given in `benchmarks_TASEPy.ipynb`.

For larger systems, the matrix $M$ is too big to solve the system of equations $MP = 0$ analytically. We thus performed stochastic simulations of the TASEP using the Gillespie algorithm and compared the local density and current to those obtained by TASEPy. The Fortran code used to simulate the TASEP is available in the repository. In Fig. 5, we compare the density profiles for $L = 50$, $\ell = 5$, $K = 4$ and $\alpha = 0.2$. The TASEPy algorithm takes less than a second on a standard laptop to compute the PSA coefficients for this medium-sized lattice.

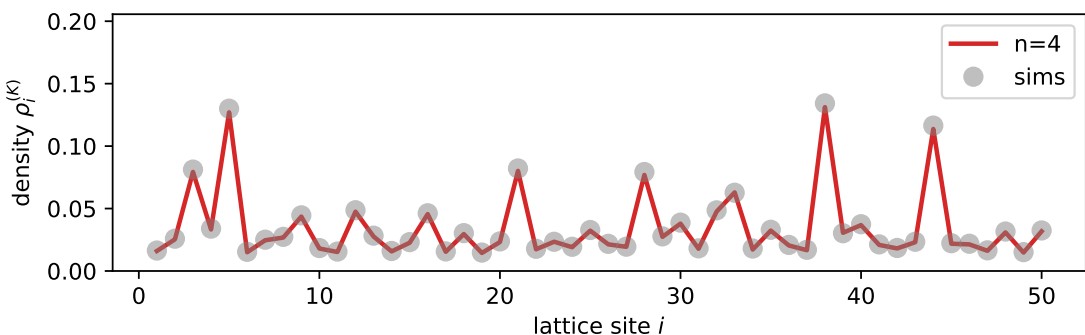

Figure 5: Comparison between density profile obtained with TASEPy (continuous lines) and stochastic simulations (gray circles) for a system with $L = 50$ and $\ell = 5$. The order of the PSA is $K = 4$ (for more details, see `benchmarks_TASEPy.ipynb`).

The simulated current has a value $J_{\mathrm{MC}} \approx 0.14262$, whereas TASEPy returns a list of currents with the percentage error $100|J^{(n)} - J_{\mathrm{MC}}|/J_{\mathrm{MC}}$ for $n = 0, \dots, K$ given by

```
TASEPy results (% error of the current):
  order 0    order 1    order 2    order 3    order 4
---------  ---------  ---------  ---------  ---------
   40.233      15.47    5.86817    2.28346   0.833051
```

The percentage error in the last order of 4 is less than 1%.

## 5   Applications

In this section, we explore two potential applications of TASEPy that are related to mRNA translation, the biological process for which the exclusion process was originally developed [1, 2]. In both applications, we use TASEPy to infer parameters of the TASEP from input data: either the mRNA-dependent translation initiation rate $\alpha$ from the mean ribosome density data obtained by polysome profiling [31], or the set of hopping rates $\omega_i$ from the local density data obtained by ribosome profiling [48].

### 5.1   Inferring initiation rates from mean density measurements

The mean ribosome density $\rho$, defined as the number of ribosomes $N$ on the mRNA divided by the length $L$ of the mRNA in codon units, can be experimentally measured with a technique

called polysome profiling. This quantity has for instance been quantified genome-wide in yeast *S. cerevisiae* [51]. As mentioned in the introduction, it is generally assumed that the codon-dependent ribosome speed mainly depends on the codon, and those rates can be roughly estimated from the abundance of the corresponding tRNA.

In Ciandrini et al. (2013) [31], the authors estimated the value of the initiation rate $\alpha$ for each mRNA of the yeast genome, assuming a codon-dependent set of ribosome hopping rates $\omega_i$. The method involved comparing the mean particle density $\rho(\alpha)$ obtained from simulated driven lattice gas to the experimental ribosome density $\rho_{\text{exp}}$. The goal was to identify the physiological initiation rate that best matches the experimentally observed densities. To achieve this, the model had to be simulated for many different values of $\alpha$ to find $\rho(\alpha) \simeq \rho_{\text{exp}}$ at an arbitrarily close resolution.

The benefit of using TASEPy lies in the fact that it eliminates the need to run numerous stochastic simulations for different values of $\alpha$ (which could be computationally expensive depending on the required points). Instead, as demonstrated in the preceding sections, our approach enables the computation of mean density coefficients $\rho_n$ to determine the mean density $\rho(\alpha)$ for any initiation rate, as seen in Eq. (25c).

In the notebook *applications_TASEPy.ipynb*, available in the repository [46], we implement this inference procedure for a random gene (YAL008W) of the *S. cerevisiae* genome. We compute the density coefficients $\rho_{i,n}$ with the `psa_compute` function, then calculate the mean density coefficients $\rho_n$, Eq. (25c), and finally use the `scipy.optimize.newton` function to find the roots of $f(\alpha) = \rho(\alpha) - \rho_{\text{exp}}$ using Newton's method since the derivative $f'$ is also computable. The initiation rate inferred with TASEPy is comparable to the one found in [31] for this particular gene ($L = 198, \ell = 9$), with a relative error of 1.5% between the two values using the third order of the PSA. The inferred value of $\alpha$ is approximately 0.15/s, which is between one and two orders of magnitude smaller than the estimated hopping rates $\omega_i$. This is a typical scenario that justifies the use of the PSA, leading to reliable results. Further details regarding the implementation of this inference procedure can be found in the notebook *applications_TASEPy.ipynb*.

## 5.2 Inferring elongation-to-initiation rate ratios from local density measurements

The exclusion process has often been used to solve the forward problem of predicting the particle current $J$ and local density $\rho_i$ from a given set of particle hopping rates $\omega_i$ for $i = 1, \dots, L$ and the initiation rate $\alpha$. In contrast, our focus lies on addressing the inverse problem, which involves inferring the rates $\omega_i$ and $\alpha$ given the local density profile $\rho_i$. Notably, as the local density $\rho_i$ is dimensionless, we can only determine the ratios $\kappa_i \equiv \omega_i/\alpha$ for $i = 1, \dots, L$, whereas the absolute rates cannot be directly inferred. Specifically, our objective is to find $\kappa_1, \dots, \kappa_L$ in a manner that satisfies the following set of conditions:

$$\rho_i(\kappa_1, \dots, \kappa_L) = \rho_{i,\text{exp}}, \quad i = 1, \dots, L, \tag{33}$$

where $\rho_{i,\text{exp}}$ represents the known local density on site $i$. This inference problem occurs when interpreting ribosome profiling experiments [52, 53], which measure local ribosome density relative to mean ribosome density, $\rho_i/\rho$ for all codons of the mRNA sequence. These data alone, however, are not enough to estimate the ratios $\kappa_i$–one needs also the mean ribosome density $\rho$ to obtain the absolute local density $\rho_i$.

For illustration purposes, we use local density obtained from stochastic simulations of the TASEP rather than from ribosome profiling. In this way, the original and inferred rates can be directly compared. In the notebook *applications_TASEPy.ipynb* [46], we demonstrate how to tackle this problem using TASEPy. The approach involves iteratively computing the PSA

coefficients and evaluating the density profiles for a given set of $\kappa_i$'s, aiming to minimize the objective function (also known as the root-mean-square deviation or RMSD):

$$\text{RMSD}(\kappa_1,\dots,\kappa_L) = \left[ \frac{\sum_{i=1}^{L}[\rho_i(\kappa_1,\dots,\kappa_L) - \rho_{i,\exp}]^2}{L} \right]^{1/2}. \tag{34}$$

The set of $\kappa_i$ for $i = 1,\dots,L$ that minimizes the RMSD will represent an optimal solution to the inverse problem.

As a proof of principle, we show this inference procedure for a small lattice ($L = 20$, $\ell = 1$), which takes few minutes on a commercial laptop ($K = 3$). The minimization of RMSD($\kappa_1,\dots,\kappa_L$) was done using `scipy.optimize.minimize` function with the Powell method and bounds on $\kappa_i$ between $10^{-2}$ and $10^5$, see *applications_TASEPy.ipynb* for more details [46]. The initial values of $\kappa_i$ were obtained using the mean-field approximation, which ignores correlations between particles [1,2]. These values were relatively close to the original values, but with obvious discrepancies (the Pearson's correlation coefficient is 0.954) [Fig. 6(a)]. After the optimization, the inferred rates provide a good match to the original ones (the Pearson's correlation coefficient is 0.998) [Fig. 6(b)]. More advanced or faster optimization methods can be implemented, but this is out of the scope of this work.

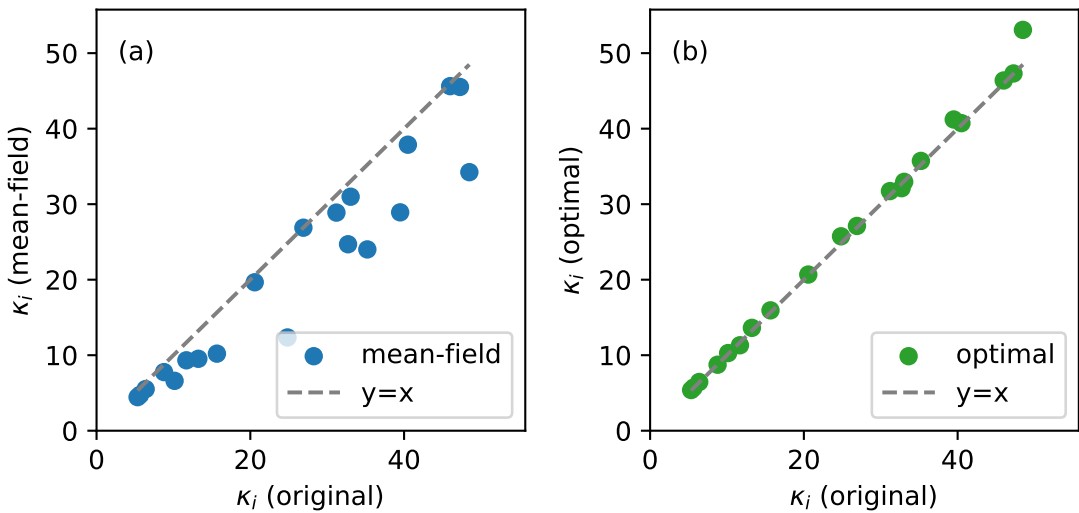

Figure 6: Results of inferring the ratios $\kappa_i = \omega_i/\alpha$ for $i = 1,\dots,L$ from the local densities. In (a) we compare the original $\kappa_i$'s (which are the ones to be estimated and used as inputs for the stochastic simulations) with their initial guess provided by the mean-field approximation of the TASEP [1,2]. Each point on the plot corresponds to one lattice site. Points that are away from the $y = x$ line indicate deviations from the original rates. After optimization (b), the inferred values of $\kappa_i$'s closely match the original ones.

We note that the inference procedure presented above is similar to the Non-Equilibrium Analysis of Ribo-seq (NEAR) procedure introduced a few years ago [48] for analysing ribosome profiling data using the inhomogeneous TASEP as a model for mRNA translation. The NEAR procedure too uses the power-series approximation of the TASEP, but is limited by construction to the third-order of the PSA ($K = 3$), whereas TASEPy has no such limitation.

# 6   Discussion and conclusion

The steady-state solution of the inhomogeneous totally asymmetric simple exclusion process (TASEP) is still unknown after more than 50 years since its first appearance. A method developed in the 1990s that provides an exact solution of the homogenous TASEP is unfortunately not applicable to the inhomogeneous TASEP, prompting a need for approximative solutions.

The power series approximation (PSA) provides an approximative solution of the inhomogeneous TASEP when one of the rates is limiting, such as the initiation rate $\alpha$ in our case. This method provides an exact series expansion of the steady-state solution in the limiting parameter, whereby the only approximation is the truncation of the series at a desired order. However, the implementation of the PSA is rather cumbersome, with results so far limited to low orders of the series expansion.

In this article, we have developed a new iterative method for computing the PSA up to any desired order $K$. We have implemented this method in a Python package called TASEPy distributed under a permissive free software licence for anyone to use. The TASEPy package computes the local density, mean density and particle current for any set of hopping rates and for any order of the PSA. Optionally, it stores the probability coefficients in a file, which is useful for analysing other physical quantities, such as density-density correlations between distance lattice sites. The correctness of the algorithm has been extensively tested using exact results for small lattice sizes and stochastic simulations.

The TASEPy iterates over all particle configurations for which the probability coefficients $c_n(C) \neq 0$ for $n \leq K$. Hence, the time complexity of the TASEPy is equal to the total number of coefficients that need to be computed. For $\ell = 1$, this number is of the order of $L^K$, where $L$ is the lattice size. Hence, it is advisable to be cautious while setting $K$ for larger lattice sizes to avoid excessive computation time. For $L = 100$ and $K = 4$, the calculation takes less than a minute on a laptop with i7 CPU and 16 GB of RAM. This indicates that TASEPy can efficiently handle computations for lattice sizes that are commonly used in literature.

We would like to emphasize that the PSA provides an approximate solution of the TASEP, which becomes increasingly accurate for larger orders $n$ and smaller values of $\alpha$. However, if the initiation rate $\alpha$ is not limiting, the approximation may not hold. This happens when the last term in the series expansion at the truncation order begins to dominate over low-order terms, leading to inaccurate or even unphysical results. This can be easily spotted by checking a number of bounds that the local density $\rho_i$ and particle current $J$ must satisfy: $0 \leq \rho_i \leq 1$, $d\rho_i/d\alpha \geq 0$, $0 \leq J \leq \alpha$ and $0 \leq dJ/d\alpha \leq 1$. Users are encouraged to use these bounds in their calculations to ensure the reliability of the results.

We also underline that, in the case $\ell = 1$, the system is particle-hole symmetric. This symmetry can be used to solve the model when $\beta$ is limiting. In practice, this can be done by replacing in the PSA equations $\alpha$ with $\beta$, $\tau_i$ with $1 - \tau_i$ for $i = 1, \ldots, L$, and $i$ with $L - i + 1$. This symmetry, however, does not hold for $\ell > 1$, and for this reason our method cannot be immediately mapped to the regime in which the exit rate is small. However, one could develop a PSA on another limiting rate ($\beta$ for instance). This and other extensions can be implemented in future versions of TASEPy.

In this paper, we have explored two practical applications of TASEPy, which are both closely related to mRNA translation, the biological process for which the TASEP was originally developed. These applications solve the inverse problem (as problems in biology often are) of inferring model inputs from model outputs: one application focuses on estimating mRNA-dependent translation initiation rate $\alpha$ from mean ribosome density measured by polysome profiling, while the other application involves determining the set of elongation-to-initiation rate ratios $\omega_i/\alpha$ from local ribosome density measured by ribosome profiling. We note, however, that these applications have been explored for illustration purposes only, and have not

been optimized for speed. Future development of TASEPy will focus on refining the provided code and optimizing the package for speed. Additionally, expanding TASEPy to other, more computationally-oriented languages will enhance its versatility and utility.

In conclusion, TASEPy provides a significant advance in solving the inhomogeneous TASEP by packaging an intricate theoretical framework into a practical, user-friendly tool that allows the exploration of the model with just a few lines of code. Due to the versatility of the TASEP, we expect the TASEPy to serve as a valuable tool for various applications. We encourage researchers to explore and enhance TASEPy further, leveraging its capabilities for their specific needs.

# Acknowledgements

**Author contributions** JSN developed the theoretical framework; JSN and LC designed and directed the project and advised RLC; LC, RLC and JSN wrote the code and performed the benchmarks. All authors discussed the results and contributed to the final manuscript.

**Funding information** LC was supported by the French National Research Agency (REF: ANR-21-CE45-0009) and by the Institut Universitaire de France (IUF). JSN was supported by a Leverhulme Trust research award (RPG-2020-327).

# A  Power Series Approximation in the case $\ell = 1$

Here we introduce the calculation of the coefficients $c_n(C)$ for the case of particles covering a single lattice site, $\ell = 1$.

We remind that the coefficients $c_n(C)$ are equal to zero if $n$ is smaller than the number of particles $N$ present in the configuration $C$, see Eq. (24). In other words, for order, $n$ we need to consider only configurations having $N \leq n$ particles. For the first order, we thus only need to compute the coefficients of configurations with one particle, which are given by Eqs. (27)-(28) as they do not depend on the number of sites covered by the particle. We report them here for clarity:

$$c_1(\{x_1\}) = \frac{1}{\omega_{x_1}}, \quad x_1 = 1, \ldots, L,$$

$$c_1(\emptyset) = -\sum_{x_1=1}^{L} \frac{1}{\omega_{x_1}}.$$

The symbol $x_1$ here stands for the position of the only particle on the lattice, and thus $\omega_{x_1}$ is the hopping rate of the particle at the position $x_1$.

For the following orders, we develop a recursive equation to compute all the coefficients

$c_n(C)$ for any order $n$ and configuration $C = \{x_1, \dots, x_N\}$ with $1 \le N \le n$.

$$c_n(\{x_1, \dots, x_N\}) = \frac{1}{e_0(\{x_1, \dots, x_N\})} \left( \underbrace{c_{n-1}(\{x_2, \dots, x_N\}) \mathbb{1}_{x_1=1}}_{(a)} \right.$$

$$+ \underbrace{\omega_{x_1-1} \mathbb{E}^1 c_n(\{x_1, \dots, x_N\}) \mathbb{1}_{x_1>1}}_{(b1)} + \underbrace{\sum_{m=2}^{N} \omega_{x_m-1} \mathbb{E}^m c_n(\{x_1, \dots, x_N\})}_{(b2)}$$

$$\left. + \underbrace{\omega_L c_n(\{x_1, \dots, x_N, L\}) \mathbb{1}_{x_N+1 \le L} \mathbb{1}_{n \ge N+1}}_{(c)} - \underbrace{c_{n-1}(\{x_1, \dots, x_N\}) \mathbb{1}_{x_1>1} \mathbb{1}_{n-1 \ge N}}_{(d)} \right),$$

where $e_0(\{x_1, \dots, x_N\})$ is given by

$$e_0(\{x_1, \dots, x_N\}) = \sum_{m=1}^{N-1} \omega_{x_m} \mathbb{1}_{x_{m+1}-x_m>1} + \omega_{x_N}.$$

Let's now go through each term (a)-(d) of the previous equation, highlighted above.

(a) This term is a contribution to $c_n(C)$ from the previous order $n-1$, provided the first particle is at site $x_1 = 1$ (condition $\mathbb{1}_{x_1=1}$).

(b) These two terms give the contribution to $c_n(\{x_1, \dots, x_N\})$ from all configurations that lead to $C = \{x_1, \dots, x_N\}$ by moving one particle to the right (provided that excluded-volume interactions allow this move). The term (b1) considers the movement of the leftmost particle (i.e. with label 1) from position $x_1 - 1$ to position $x_1$ (and thus one needs to check that $x_1 > 1$). The term (b2) considers the stepping of all other particles $m = 2, \dots, N$ from $x_m - 1$ to $x_m$, and for which one does not have to check the condition $x_1 > 1$.

(c) The case in which a particle exits the lattice is considered in (c). This reduces the number of particles from $N + 1$ to $N$, resulting in the configuration $C = \{x_1, \dots, x_N\}$. Since this coefficient is equal to zero if $n$ is smaller than the number of particles $N$ in the configuration $C$, one needs to check that $N + 1 \le n$. Furthermore, because of volume exclusion, the position $x_N$ must less than or equal to $L - 1$, i.e. the condition $x_N + 1 \le L$ has to be satisfied.

(d) The last term (d) removes the contribution of the same configuration $C$ from the previous order $n - 1$, provided that the first particle is not on the first site ($x_1 > 1$). This term comes from exiting $C$ by means of adding a new particle at the lattice site 1, which can occur only if $x_1 > 1$. As above, $n - 1 \ge N$ otherwise this term is zero.

The coefficients $c_n$ can be computed recursively following a precise order of configurations that allows evaluating $c_n$ such that the terms (a)-(d) are known. To explain what this order of configurations is, let us imagine that all $c_{n-1}$ have been computed, and we want to calculate the coefficients $c_n$. We first consider the case $n \le L$. The first configuration we need to consider is the one with $n$ particles stacked at the beginning of the lattice (we remind that $c_n(C) = 0$ for any $C$ with more particles than $n$). For $\ell = 1$, this 'stacked' configuration is simply $\{1, 2, \dots, n\}$. We can then compute the coefficient of order $n$ for this configuration, as only the term (a) contributes:

$$c_n(\{1, 2, \dots, n\}) = \frac{1}{\omega_{x_n}} c_{n-1}(\{2, \dots, n\}).$$

Next, we compute the coefficient for the configuration in which the $n$th particle is moved one step to the right ($x_n = n + 1$). Importantly, this coefficient depends only on the previously computed coefficient $c_n(\{1, 2, \ldots, n\})$ via the term (b2), and on other coefficients that are of order $n-1$, which are known. The terms (b1) and (c) are zero since the first site is occupied and the condition $n \geq N + 1$ cannot be satisfied. This procedure can be iterated until the $n$th particle reaches the site $L$. From there, the next configuration to be computed is the one obtained by removing the $n$th particle; the corresponding coefficient can be computed since the contribution from (c) is known.

We now have all the coefficients needed to compute the coefficient having the first $n - 2$ particles stacked together, with the $(n-1)$th particle moved by one lattice site to the right, and the $n$th particle stacked next to it (the configuration $\{1, 2, \ldots, n-2, n, n+1\}$. We then cycle through all configurations keeping the $(n-1)$th particle fixed, while moving the $n$th particle step by step, including the configuration in which the $n$th particle exits the lattice.

From there, the next configuration is $\{1, 2, \ldots, n-2, n+1, n+2\}$. We repeat this procedure until we visit the configuration with both the $(n-1)$th and the $n$th particles removed from the lattice. The next configuration to be computed is the one with the $(n-2)$th particle moved one site to the right, and the other two ($n-1$ and $n$th) stacked next to it. These steps are repeated over and over, until we obtain a configuration in which there is only one particle left, residing on the last site, $x_1 = L$. The next configuration would be the empty one, whose coefficient $c_n(\emptyset)$ can be obtained from Eq. (14).

If instead we need to compute the coefficients of an order larger than the lattice size, $n > L$, we start from a stacked configuration with particles occupying all the sites of the lattice, and go through the same procedure as above until we obtain the empty configuration. Thus, if $n \geq L$, all the configurations are explored with this procedure, whereas if $n < L$ only a subset of configurations is visited.

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
