# Peer review of "TASEPy: a Python-based package to iteratively solve the inhomogeneous exclusion process"

_SciPost Physics Codebases_

## Round 2 · Referee Report · Anonymous (Referee 2) · 2023-9-20

Strengths

1) The article provides and describes a Python package to find the steady state of the inhomogeneous TASEP. The method relies on a power series approximation, consisting in expanding the solution of the master equation in powers of a (slow) injection rate. It provides the first systematic implementation of this method.

2) The manuscript is clear and well written. The general introduction, the presentation of the model and the presentation of the method are very pedagogical and accessible to a large audience.

3) The description of the functions included in the package are clear and make the package easy to use.

4) The implementation is successfully tested against both (i) exact solutions for small lattice size and (ii) direct numerical simulations for larger systems. Several applications and tests are provided.

Weaknesses

The current implementation is restricted to the computation of two quantities: the mean density of particles and the mean current. I think the article and the Python package could gain much more impact with a few modifications that would allow the computation of any physical quantity.

Report

I think this manuscript deserves publication, but I feel that it could gain significant impact with little modification to the algorithm.

The power series approximation used by the authors gives a perturbative solution of the master equation (in the steady state). This solution contains all the physical information about the steady state of the system. The implementation of the algorithm computes this solution, order by order, but does not store the result. Instead, it is used, at each order, to compute the physical quantities of interest here (current and density). This is a bit of a shame, as this full solution could be of interest to study any physical observable, and not just the two considered here (for instance density-density correlations between different sites).
My guess is that this result is not stored for memory issues, as the full probability of configurations for a large lattice would represent too much data (but this is not stated in the manuscript). On the other hand, the simplification (24) identified by the authors should greatly reduce the amount of memory needed for storing this result, so that might be feasible.

Requested changes

1) Could the authors add a function that gives the full solution of the master equation up to a given order, so that it could be used to study other quantities? If this is not feasible, maybe the function psa_compute could be modified to take as an argument a list of quantities to compute from the master equation (these quantities would be functions of the list of occupation numbers)? This second option would boil down to changing Eq. (25) to use instead of (3) or (4) any function of the occupation numbers $\tau_i$.
I feel that this would significantly increase the impact of the article, with relatively low effort. If, for some technical reason that I missed this is not possible, it would be worth adding a comment in the manuscript about this point.

Minor comments/suggestions:
2) In Eq. (2b) and (2c), perhaps it would be more clear to write the evolution of the pair of occupation numbers using brackets, such as $ (\tau_i, \tau_{1+1}) = (1,0) \rightarrow (0,1)$.
3) For Eq. (3) a quick explanation of the formula would be useful. For instance, to explain that the current from the left can be written as the injection rate multiplied by the probability that all the sites between $0$ and $l$ are empty. Equivalently, it is equal to the exit current from the right, which can be written as the exit rate multiplied by the probability that the last site is occupied.
4) There are two references missing in the first paragraph of Section 5, which currently appear as "[]".

  • validity: top
  • significance: good
  • originality: high
  • clarity: top
  • formatting: excellent
  • grammar: excellent

Author:  Luca Ciandrini  on 2023-10-31  [id 4084]

(in reply to Report 2 on 2023-09-20)

We thank the reviewer for their positive comments and suggestions.

R: [...] 1) Could the authors add a function that gives the full solution of the master equation up to a given order, so that it could be used to study other quantities? [...]
A: In 'psa_compute' we indeed compute all the coefficients, but we do not store them. We have now added this feature by means of two optional arguments in 'psa_compute', 'save_coeffs=True' (the default value is False if omitted), and 'coeffs_file=$\langle \textit{filename}\rangle$', which specifies the file for storing the coefficients. The coefficients are stored in one row per coefficient, in the format $n$ (order), $X$ (list of particle positions) and $c_n(X)$ (the coefficient). Given that the number of coefficients scales exponentially with $L$, and therefore the resulting file may be large, we have implemented another function called 'total_coeffs', which computes the total number of coefficients that will be stored in the file. This number can be used to decide whether storing the coefficients is feasible. For example, for $L=100$, $\ell=1$ and $K=4$, 'total_coeffs' returns circa 4,300,000 coefficients, which take about 173 MB of space, or about 43 bytes per coefficient. We have changed the text on pages 10 and 11 to explain these two new optional arguments, and updated the tutorial showing these new features in practice.

The alternative option of passing a function of lattice occupancy variables will be added in future versions of TASEPy.

R: 2) In Eq. (2b) and (2c), perhaps it would be more clear to write the evolution of the pair of occupation numbers using brackets, [...]
A: We have changed this in the text.

R: 3) For Eq. (3) a quick explanation of the formula would be useful. [...]
A: We have added the following sentence in the text: ``As the steady-state current is conserved along the lattice, the equation above can be understood as equating the entry current (injection rate $\alpha$ multiplied by the probability that the first $\ell$ sites are empty), and the exit current on the last site (termination rate $\beta$ multiplied by the probability of occupancy of the last site)."

R: 4) There are two references missing in the first paragraph of Section 5, which currently appear as "[]".
A: We have fixed the references.

---

## Round 2 · Referee Report · Anonymous (Referee 1) · 2023-9-20

Strengths

1. Addresses important problem
2. Useful for most researchers in the field
3. Scientifically sound
4. Software is easy to use

Weaknesses

1 . Presentation could be improved, e.g. by considering the special case l=1 first.
2. Definition of "tracking site" could be improved.
3. Discussion of symmetries of ASEP could be useful.

Report

In the manuscript the authors developed and describe a Python-based package that allows to solve general inhomogeneous exclusion processes in one dimension exactly on finite lattices. Much of the theory behind the proposed expansion has already been discussed, mainly by the authors themselves, in previous publications. Here the focus is more on the functionalities and usage of the Python package.

Despite the limitations on system size and/or order of the expansion, the presented tool is extremely helpful for any researcher in the field of 1d stochastic processes. It might help to trigger new conjectures for
analytical results which then could be proved using mathematical methods. Therefore, in principle, I recommend publication. The acceptance criteria of SciPost Physics Codebases are fulfilled.

However, there might be a few points that the authors want to consider.

Pedagogically it might be better to present the algorithm for l=1 first.
This would avoid some complications in the notation that could be confusing at first, e.g. the introduction of a "tracking site" (in Fig.1), which is not otimally explained in the text. The generalization to general l seems than to be rather straightforward. If necessary, some aspects could be explained in a subsection or an Appendix.

The expansion is in the input parameter alpha which needs to be small. Can certain symmetries of the ASEP be used to extend the validity to other parameter values, e.g. small beta? A discussion of such symmetries might be helpful for the reader.

Requested changes

1. Optional: consider focus on case l=1
2. Discuss symmetries of ASEP and its potential usage.

  • validity: top
  • significance: top
  • originality: high
  • clarity: good
  • formatting: excellent
  • grammar: excellent

Author:  Luca Ciandrini  on 2023-10-31  [id 4083]

(in reply to Report 1 on 2023-09-20)

We would like to thank the reviewer for their positive comments and remarks.

Below we reply to the major points that have been raised.

R: Pedagogically it might be better to present the algorithm for $\ell=1$ first. [...]
A: We have now added a more pedagogical section to better introduce the reader to the $\ell$ TASEP and the tracking site. We did not want to modify too much the flow of the main text, so this section is added as an Appendix and referenced in the main text.

We have also added the following sentence at page 3 to better explain the ``tracking site":
"In order to locate each particle on the lattice, we arbitrarily chose a site of the particle that we will call the ``tracking site'' (identical for all particles); we can then identify the position of a particle on the lattice by the position of its ``tracking" site. For instance, in Fig.1 each particle has size $\ell =3$ and the tracking site is the middle one. "

R: [...] Can certain symmetries of the ASEP be used to extend the validity to other parameter values, e.g. small beta? [...]
A: For $\ell=1$, there is indeed a symmetry that can be used to solve the model when the parameter $\beta$ is small. This symmetry amounts to replacing $\tau_i\leftrightarrow 1-\tau_i$ for $i=1,\dots,L$, $\alpha\leftrightarrow\beta$ and $i\leftrightarrow L-i+1$. This symmetry, however, does not hold for $\ell>1$, and for this reason our method cannot be immediately mapped to the regime in which the exit rate is small.
We have commented this in the Discussion section. That said, we note that the power-series expansion can be developed for small $\beta$ for $\ell>1$, and we plan to add this feature in future versions of the TASEPy package.

---

## Editorial Decision

resubmitted